# Sex-based differences in clearance of chronic *Plasmodium falciparum* infection

**Jessica Briggs[1]\*, Noam Teyssier[1], Joaniter I Nankabirwa[2,3], John Rek[2], Prasanna Jagannathan[4], Emmanuel Arinaitwe[2,5], Teun Bousema[6,7], Chris Drakeley[7], Margaret Murray[4], Emily Crawford[8], Nicholas Hathaway[9], Sarah G Staedke[5], David Smith[10], Phillip J Rosenthal[1], Moses Kamya[2,3], Grant Dorsey[1], Isabel Rodriguez-Barraquer[1†], Bryan Greenhouse[1†]**

[1]Department of Medicine, University of California San Francisco, San Francisco, United States; [2]Infectious Diseases Research Collaboration, Kampala, Uganda; [3]Department of Medicine, Makerere University College of Health Sciences, Kampala, Uganda; [4]Department of Medicine, Stanford University, Palo Alto, United States; [5]Department of Clinical Research, London School of Hygiene and Tropical Medicine, London, United Kingdom; [6]Department of Medical Microbiology, Radboud University Nijmegen Medical Centre, Nijmegen, Netherlands; [7]Department of Immunology and Infection, London School of Hygiene and Tropical Medicine, London, United Kingdom; [8]Chan-Zuckerberg Biohub, San Francisco, United States; [9]Department of Medicine, University of Massachusetts, Amherst, United States; [10]Institute for Health Metrics & Evaluation, University of Washington, Seattle, United States

**\*For correspondence:**
Jessica.Briggs@ucsf.edu

[†]These authors contributed equally to this work

**Abstract** Multiple studies have reported a male bias in incidence and/or prevalence of malaria infection in males compared to females. To test the hypothesis that sex-based differences in host-parasite interactions affect the epidemiology of malaria, we intensively followed *Plasmodium falciparum* infections in a cohort in a malaria endemic area of eastern Uganda and estimated both force of infection (FOI) and rate of clearance using amplicon deep-sequencing. We found no evidence of differences in behavioral risk factors, incidence of malaria, or FOI by sex. In contrast, females cleared asymptomatic infections at a faster rate than males (hazard ratio [HR]=1.82, 95% CI 1.20 to 2.75 by clone and HR = 2.07, 95% CI 1.24 to 3.47 by infection event) in multivariate models adjusted for age, timing of infection onset, and parasite density. These findings implicate biological sex-based differences as an important factor in the host response to this globally important pathogen.

## Introduction

Malaria, a protozoan infection of the red blood cells, remains one of the greatest global health challenges (*World malaria report, 2019*). Infection with malaria parasites results in a wide range of clinical disease presentations, from severe to uncomplicated; in addition, in hyperendemic areas, asymptomatic infections are common (*Bousema et al., 2014*). It is well established that chronic asymptomatic infection with *Plasmodium falciparum*, the most common and fatal malaria parasite, can lead to morbidity for those infected and contribute to ongoing transmission (*Bousema et al., 2014*; *Okell et al., 2012*; *Tadesse et al., 2018*; *Slater et al., 2019*). Characterization of these asymptomatic infections is paramount as they represent a major obstacle for malaria elimination efforts. Therefore, an understanding of how host immunity and parasite factors interact to cause disease tolerance is required. While age-specific immunity to malaria in hyperendemic areas is well-

characterized, less attention has been paid to the possibility of a sex bias in malarial susceptibility despite evidence for a male bias in malaria infections in non-human animals and a male bias in the prevalence of other human parasitic infections (*Zuk and McKean, 1996*; *Klein, 2004*; *Roberts et al., 2001*).

The clearest evidence for sexual dimorphism in malaria susceptibility is in pregnant women, who are at greater risk of malaria infection and also experience more severe disease and higher mortality (*Desai et al., 2007*). However, multiple studies performed in different contexts have demonstrated a male bias in incidence and/or prevalence of malaria infection in school-aged children and adults (*Molineaux et al., 1980*; *Pathak et al., 2012*; *Landgraf et al., 1994*; *Camargo et al., 1996*; *Abdalla et al., 2007*); this bias is more well established in hypoendemic areas, but has also been observed in hyperendemic regions (*Houngbedji et al., 2015*; *Mulu et al., 2013*). Where there is a male bias in malaria infection, it has often been postulated that these differences in malaria incidence or prevalence stem from an increased risk of males acquiring infection due to socio-behavioral factors (*Pathak et al., 2012*; *Camargo et al., 1996*; *Moon and Cho, 2001*; *Finda et al., 2019*). However, because biological sex itself has been demonstrated to affect responses to other pathogens, an alternative hypothesis is that the sexes may have different responses to the malaria parasite once infected (*Nhamoyebonde and Leslie, 2014*; *Bernin and Lotter, 2014*; *Fischer et al., 2015*; *Fish, 2008*).

Estimating the host response to *P. falciparum* infection requires close follow-up of infected individuals, sensitive detection of parasites, and the ability to distinguish superinfection, which is common in endemic areas, from persistent infection. To test the hypothesis that sex-based differences in host–parasite interactions affect the epidemiology of malaria, we intensively followed a representative cohort of individuals living in a malaria endemic area of eastern Uganda. Using frequent sampling, ultrasensitive quantitative PCR (qPCR), and amplicon deep-sequencing to genotype parasite clones, we were able to accurately detect the onset of new infections and follow all infections over time to estimate their duration. Using these data, we show that females cleared their asymptomatic infections more rapidly than males, implicating biological sex-based differences as important in the host response to this globally important pathogen.

## Results

### Cohort participants and *P. falciparum* infections

This analysis involved data from 477 children and adults (233 males and 244 females) that were followed for a total of 669.6 person-years (*Table 1*). 25 participants, 10 males and 15 females, were enrolled after initial enrollment (*Figure 1*); median duration of follow-up in those who were dynamically enrolled was 0.84 years (IQR 0.69–1.23) compared to 1.45 years (IQR 1.43–1.47) in those enrolled during initial enrollment. 149 of 477 participants (31.2%) included in the analysis had at least one *P. falciparum* infection detected (*Figure 1*). 114 participants had 822 successfully genotyped samples and had infections characterized by clone and by infection event. 35 samples (from 35 unique participants) had very low-density infections (<1 parasite/µL) that could not be genotyped; these infections were characterized at the infection event level only. We achieved a read count of >10,000 for 92% of genotyped samples, identifying 45 unique AMA-1 clones in our population (frequencies and sequences in *Supplementary file 1e*). At the clone level, the proportion of baseline infections out of all infections in males was 117/171 (68.4%), compared to 68/116 (58.8%) baseline infections in females (p=0.10). At the infection event level, there were 54/104 (51.9%) baseline infections in males and 45/89 (50.6%) baseline infections in females (p=0.89).

### Behavioral malaria risk factors and measures of malaria burden

There was no difference in reported rates of LLIN use the previous night by sex (*Table 1*). Women over the age of 16 traveled overnight outside of the study area more than men (incidence rate ratio [IRR] for females vs. males = 3.61, 95% confidence interval [CI] 1.83 to 7.13), a potential risk factor for malaria exposure. Antimalarial use outside the study clinic was reported only four times (3 females and one male, all under the age of 5). In this region receiving regular rounds of IRS, the incidence of symptomatic malaria was low in all age categories, and there was no evidence of a difference in incidence of symptomatic malaria by sex overall (IRR for females vs. males = 1.11, 95% CI

**Table 1.** Behavioral risk factors for malaria infection and measures of malaria burden in study population, stratified by age and sex.

| Metric | Age and gender categories | | | | | | | |
|---|---|---|---|---|---|---|---|---|
| | All | | <5 years old | | 5–15 years old | | 16 years and older | |
| | Male | Female | Male | Female | Male | Female | Male | Female |
| Number of participants, n | 233 | 244 | 73 | 84 | 101 | 71 | 59 | 89 |
| Median days of follow-up per participant | 530.0 | 530.0 | 525.0 | 524.5 | 530.0 | 531.0 | 530.0 | 530.0 |
| Slept under LLIN the previous night | 53.6% | 56.3% | 54.1% | 56.3% | 47.9% | 47.8% | 63.7% | 64.3% |
| Person-years of follow-up | 324.2 | 345.4 | 87.3 | 96.7 | 152.6 | 118.5 | 84.22 | 130.1 |
| Number of overnight trips | 44 | 107 | 21 | 19 | 9 | 10 | 14 | 78 |
| Incidence of overnight trips*, (95% CI) | 0.14 (0.09–0.20) | 0.31 (0.20–0.49) | 0.24 (0.14–0.42) | 0.20 (0.09–0.42) | 0.06 (0.03–0.13) | 0.08 (0.03–0.24) | 0.17 (0.09–0.31) | 0.60 (0.30–1.18) |
| Episodes of malaria* | 11 | 13 | 5 | 2 | 5 | 9 | 1 | 2 |
| Incidence of malaria**, (95% CI) | 0.03 (0.02–0.06) | 0.04 (0.02–0.09) | 0.06 (0.02—0.16) | 0.02 (0.00–0.11) | 0.03 (0.01–0.08) | 0.08 (0.03–0.23) | 0.01 (0.00–0.08) | 0.02 (0.00–0.17) |
| Number of routine visits, n | 4316 | 4583 | 1164 | 1293 | 2034 | 1568 | 1118 | 1722 |
| Prevalence of microscopic parasitemia*** | 2.9% | 1.4% | 1.8% | 1.1% | 4.4% | 2.7% | 1.3% | 0.5% |
| Prevalence of parasitemia by qPCR | 14.4% | 9.2% | 5.8% | 3.7% | 17.0% | 15.3% | 18.5% | 7.7% |
| Geometric mean parasite density**** | 3.41 | 3.06 | 4.86 | 13.06 | 6.31 | 4.20 | 1.09 | 1.02 |
| Median complexity of infection, (IQR) | 3 (1–7) | 2 (1–2) | 1 (1–2.5) | 2 (1–2.3) | 4 (2–9) | 2 (1–2) | 1 (1–2) | 1 (1–2) |

*Malaria includes one episode (female,<5 years old), due to non-falciparum species (*P. malariae*).

**per person-year.

***Parasitemia by light microscopy includes one episode (female, 5–15 years old) due to non-falciparum species (*P. ovale*).

****Geometric mean parasite density in parasites/μL of all qPCR-positive routine visits.

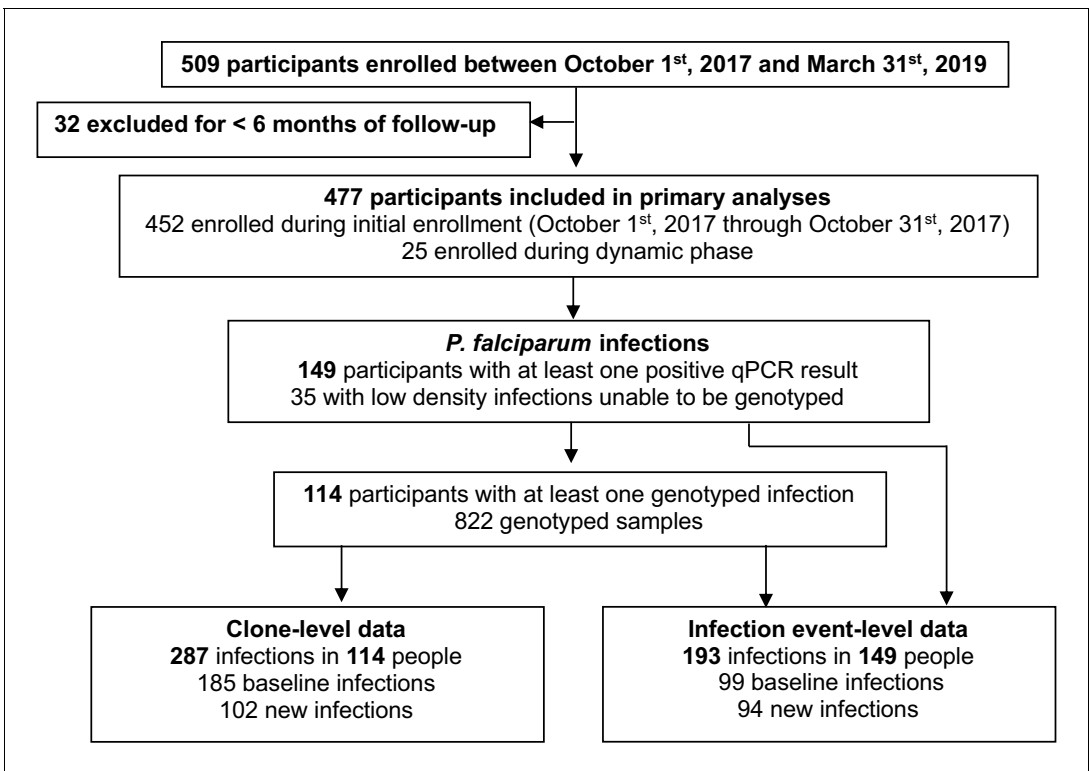

**Figure 1.** Study design.

0.49 to 2.52) or when adjusted for age (IRR = 1.26, 95% CI 0.54 to 2.96). In contrast, prevalence ratio (PR) of *P. falciparum* parasitemia by microscopy in females versus males across all age categories was 0.49 (95% CI 0.36 to 0.65), with relative differences in prevalence most pronounced in the oldest age group. Similar findings were seen when prevalence was assessed by ultrasensitive qPCR, with PR = 0.64 in females vs. males (95% CI 0.43 to 0.96), again with the largest differences seen in the oldest age group. Adjusting for age as a categorical variable, LLIN use, and travel did not qualitatively change prevalence ratios for microscopic parasitemia (PR in females vs. males = 0.57, 95% CI 0.42 to 0.77) or for qPCR-positive parasitemia (PR in females vs. males = 0.67, 95% CI 0.60 to 0.76). We also found no evidence for a difference in parasite density as determined by qPCR between males and females after adjusting for age as a continuous variable (p=0.47). Median complexity of infection (COI) was higher in males than in females overall, driven primarily by a higher COI in male school-aged children.

## Force of infection by age and sex

To determine whether higher infection prevalence in males was due to an increased rate of infection, we used longitudinal genotyping to calculate the force of infection (FOI, number of new blood stage infections per unit time). Overall, the FOI was low, with new infections occurring on average less than once every 5 years (*Table 2*). There was no evidence for a significant difference in FOI by sex overall (IRR for females vs. males = 0.88, 95% CI 0.48 to 1.62 by clone and IRR = 0.83, 95% CI 0.52 to 1.33 by infection event). There was also no evidence for a significant difference in FOI by sex when adjusted for age category (IRR = 0.88, 95% CI 0.47 to 1.63 by clone and IRR = 0.83, 95% CI 0.53 to 1.31 by infection event). For analysis both by clone and by infection event, there was a trend toward higher FOI in males compared to females. We performed a sensitivity analysis by decreasing the number of skips necessary to declare an infection cleared (*Supplementary file 1f*). Performing the same analysis using one skip for clearance instead of 3 skips increased the FOI in all groups and increased the trend toward higher FOI in males, with IRR for females vs. males when adjusted for age category = 0.71, 95% CI 0.41 to 1.24 by clone and IRR = 0.75, 95% CI 0.47 to 1.22 by infection event.

## Rate of clearance of infection and duration of infection by sex

Since females had a lower prevalence of infection but similar rate of acquiring infections compared to males, we evaluated whether there was a difference between sexes in the rate at which infections were cleared. Asymptomatic infections were included in this analysis if they were not censored as stated in the methods. At the clone level, 105 baseline infections and 53 new infections were included; there was a slightly higher proportion of baseline infections in males (68/99, 68.7%), compared to the proportion of baseline infections in females (37/59, 62.7%) (p=0.49). At the infection event level, 58 baseline infections and 51 new infections were included and there was no difference in the proportion of baseline infections by sex, with 32/60 (53.3%) baseline infections in males and 26/49 (53.1%) baseline infections in females (p=1.0).

Unadjusted hazard ratios for clearing infecting clones showed that asymptomatic infections cleared naturally (*i.e.,* when not treated by antimalarials) at nearly twice the rate in females vs. males (hazard ratio (HR) 1.92, 95% CI 1.19 to 3.11, *Table 3*). In addition, new infections cleared faster than

**Table 2.** Molecular force of infection (FOI) by clone and by infection event, stratified by age and sex.

| Molecular force of infection (FOI) | Sex | Age category | | | |
| | | All | <5 years | 5–15 years | 16 years or older |
|---|---|---|---|---|---|
| By clone, ppy* (95% CI) | All | 0.18 (0.13–0.24) | 0.14 (0.07–0.28) | 0.19 (0.08–0.43) | 0.20 (0.08–0.46) |
| | Male | 0.19 (0.12–0.30) | 0.16 (0.07–0.39) | 0.19 (0.10–0.37) | 0.22 (0.09–0.54) |
| | Female | 0.17 (0.09–0.31) | 0.12 (0.03–0.49) | 0.19 (0.08–0.45) | 0.18 (0.06–0.54) |
| By event, ppy* (95% CI) | All | 0.14 (0.11–0.18) | 0.09 (0.06–0.16) | 0.16 (0.09–0.30) | 0.16 (0.08–0.32) |
| | Male | 0.16 (0.11–0.22) | 0.13 (0.07–0.27) | 0.18 (0.11–0.28) | 0.15 (0.07–0.29) |
| | Female | 0.13 (0.08–0.21) | 0.06 (0.02–0.18) | 0.14 (0.07–0.27) | 0.17 (0.07–0.41) |

*per person-year.

**Table 3.** Hazard ratios for rates of clearance of infection, by clone and by infection event.

| Predictors | Categories | Hazard ratio by clone (95% CI) | | Hazard ratio by infection event (95% CI) | |
|---|---|---|---|---|---|
| | | Unadjusted | Adjusted | Unadjusted | Adjusted |
| Sex | Male | ref | ref | ref | ref |
| | Female | 1.92 (1.19–3.11) | 1.82 (1.20–2.75) | 2.30 (1.20–4.42) | 2.07 (1.24–3.47) |
| Age | 16 years or greater | ref | ref | ref | ref |
| | 5–15 years | 0.66 (0.39–1.10) | 0.81 (0.49–1.36) | 0.82 (0.39–1.74) | 1.27 (0.72–2.25) |
| | <5 years | 1.64 (0.79–3.41) | 1.55 (0.76–3.17) | 2.01 (0.80–5.00) | 1.75 (0.87–3.53) |
| Complexity of infection (COI) | Polyclonal (COI > 1) | ref | – | ref | – |
| | Monoclonal (COI = 1) | 1.63 (1.03–2.57) | – | 0.95 (0.38–2.34) | – |
| Infection status | Present at baseline | ref | ref | ref | ref |
| | New infection | 1.94 (1.22–3.07) | 1.75 (1.05–2.94) | 4.66 (2.58–8.42) | 4.32 (2.59–7.20) |
| Parasite density * | | 0.85 (0.69–1.06) | 0.81 (0.65–1.00) | 0.41 (0.32–0.51) | 0.44 (0.35–0.54) |

*Increasing parasite density (log10) in parasites/microliter, as measured by qPCR.

baseline infections and monoclonal infections cleared faster than polyclonal infections. Unadjusted hazard ratios for clearance of infection events (as opposed to clones) also showed faster clearance in females vs. males (HR = 2.30, 95% CI 1.20 to 4.42). Results were similar in multivariate models including age, gender, the period during which the infection was first observed, and parasite density, demonstrating faster clearance in females vs. males (HR = 1.82, 95% CI 1.20 to 2.75 by clone and HR = 2.07, 95% CI 1.24 to 3.47 by infection event). Complexity of infection was not included in the final adjusted model because the model fit the data less well when COI was included as a predictor. In both adjusted models, new infections cleared faster than baseline infections. Higher parasite densities were associated with slower clearance by clone and by infection event, but the effect size was larger when data were analyzed by infection event (HR = 0.44, 95% CI 0.35 to 0.54). There was no evidence for interaction between age and sex in either adjusted model. We performed a sensitivity analysis by decreasing the number of skips necessary to declare an infection cleared. Regardless of whether three skips, two skips, or one skip was used to determine infection clearance, females cleared their infections faster than males both by clone and by infection event (*Supplementary file 1g*).

We next estimated durations of asymptomatic infection by age and sex using results from a model that included these covariates (*Figure 2*). Durations of infection ranged from 103 days to 447 days by clone, and from 87 to 536 days by infection event. Males had a longer duration of infection across all age categories. Children aged 5–15 years had the longest duration of infection, followed by adults. Therefore, overall, males aged 5–15 years had the longest estimated duration of infection by either clone (447 days) or infection event (526 days).

## Discussion

Previous studies have reported a higher prevalence of malaria infection in males compared to females, with the difference often ascribed to differences in exposure (*Molineaux et al., 1980*; *Pathak et al., 2012*; *Landgraf et al., 1994*; *Camargo et al., 1996*; *Abdalla et al., 2007*; *Houngbedji et al., 2015*; *Mulu et al., 2013*). In a cohort study in eastern Uganda, we noted higher prevalence of malaria infection in males compared to females. By closely following a cohort of children and adults and genotyping every detected infection with sensitive amplicon deep-sequencing, we were able to estimate both the rate of infection (FOI) and duration of infection and to compare these measures by sex. We found that lower prevalence in females did not appear to be due to lower rates of infection but rather due to faster clearance of asymptomatic infections. To our knowledge, this is the first study to report a sex-based difference in the duration of malaria infection.

Though there are some conflicting reports in the literature, the majority of studies of malaria incidence and/or prevalence that evaluated associations with sex in late childhood, adolescence and

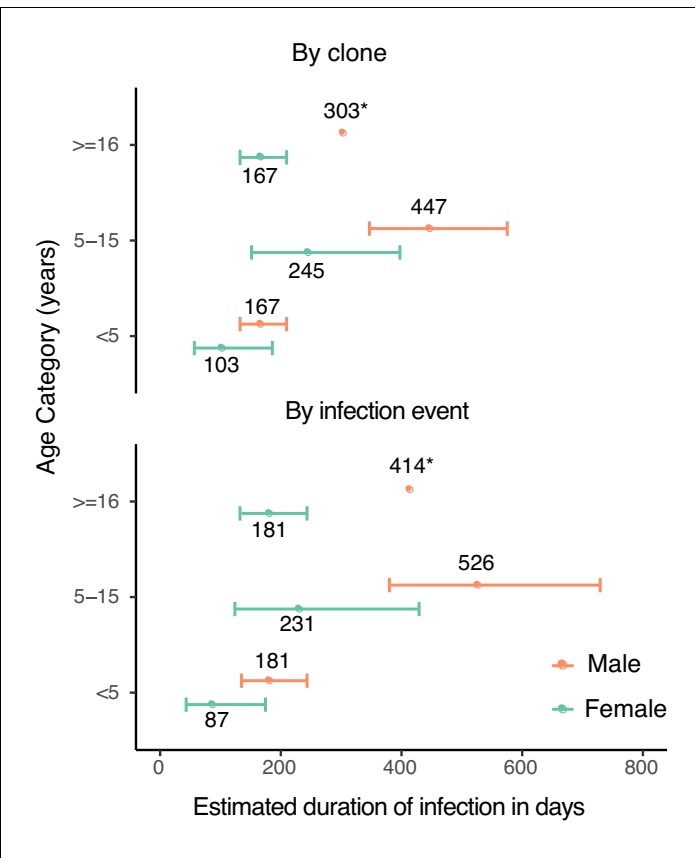

**Figure 2.** Estimates of duration of infection from sex- and age-adjusted model. Estimated duration of infection in days, calculated by adjusting the point estimate of the baseline hazard by the coefficients of the sex- and age-adjusted model. Error bars represent standard errors of duration obtained from variance in the model coefficients. Point estimates of duration are labeled (*).

adulthood have found a male bias in the observed measure of burden (*Molineaux et al., 1980*; *Pathak et al., 2012*; *Landgraf et al., 1994*; *Camargo et al., 1996*; *Abdalla et al., 2007*; *Houngbedji et al., 2015*; *Mulu et al., 2013*). We note that this is more often observed in hypoendemic settings and may be confounded by factors such as treatment-seeking behavior; however, this male bias has been reported in studies of both *P. vivax* and *P. falciparum*. Overall, these studies consistently suggest that males exhibit higher incidence and/or prevalence of malaria that begins during late childhood, persisting through puberty and the majority of adulthood (excepting the years when pregnancy puts women at higher risk). One possible explanation put forward for the sex-specific difference in burden has been that males are more frequently bitten by malaria-carrying mosquitos due to behavioral differences such as working outside, not sleeping under a net, or traveling for work (*Pathak et al., 2012*; *Camargo et al., 1996*; *Moon and Cho, 2001*; *Finda et al., 2019*). In our study, however, there were no statistically significant differences in malaria incidence or FOI by sex, though there was a trend toward higher FOI in males. We also saw no evidence of behavioral trends that would result in more infections in males; in fact, older women in our study did most of the traveling outside the study area (an area of low transmission compared to surrounding areas). We did not assess work habits as part of our study questionnaire, but the fact that we observe similar patterns in prevalence by sex in all age categories makes this a less likely explanation. Therefore, in this particular cohort, it appears that the observed male bias in parasite prevalence is best explained by slower clearance of infection in males. We cannot rule out that males were differentially exposed in the recent past (e.g., had a higher force of infection when transmission was higher, which could have affected immunity), and we plan to genotype samples from a prior cohort from 2011 to 2017 to test this hypothesis.

Very few studies have been conducted to explore immunological differences between males and females in their response to the malaria parasite. RTS,S vaccination is associated with higher all-cause mortality in girls compared to boys, and a trend toward higher risk of fatal malaria has been noted in vaccinated girls compared to boys, suggesting possible sexual dimorphism in immunological responses to malaria (*Klein et al., 2016*). The comprehensive Garki project found a male bias in the prevalence of *P. falciparum* infection after the age of 5 years that was noted to increase after control measures including IRS and mass drug administration. They also found that females had higher levels of certain antibodies against *P. falciparum* compared to males, but saw no difference in these levels after control measures (*Molineaux et al., 1980*). Hormonal differences have been posited to play a role in these sex-based immunological differences to malaria infection; for example, studies in mice show that testosterone appears to downregulate the immune response to malaria (*Delić et al., 2011*; *Wunderlich et al., 1991*). One study of irradiated sporozoite vaccination in mice showed that female mice vaccinated after puberty were better protected than males following parasite challenge, but this sex difference was not seen if the mice were vaccinated prior to puberty, suggesting a role for sex hormones in the development of immunity (*Vom Steeg et al., 2019*). Testosterone was again associated with decreased protection from malaria in that study (*Vom Steeg et al., 2019*). In addition, in Kenya, two studies showed that dehydroepiandrosterone sulfate (DHEAS) levels were significantly associated with decreased parasite density in both males and females, even after adjustment for age, but neither study directly compared males to females (*Kurtis et al., 2001*; *Leenstra et al., 2003*). Given that levels and types of sex hormones differ between males and females, as does the age of onset of puberty, the interaction between these hormones, age, and protection against malaria is likely to be complicated. We did not detect an interaction between age and sex in our model. This could be due to a lack of statistical power or imperfect detectability, as adults have lower density infections than children due to improved anti-parasite immunity (*Bousema et al., 2014*; *Okell et al., 2012*). In addition, the adult age group was comprised of a wide age range, which included post-menopausal women and might have blunted the effect of sex in that age group if there is a hormonal basis for some of this effect. Repeating our analysis using different age categories (<8 years, 8–13 years, and 13+ years) did not significantly change our findings. It is also possible that sex-based differences are explained by a different mechanism, such as differences in innate immunity, Y-linked chromosomal factors in males, increased X-linked gene expression in females, or by a combination of factors. More studies are needed to elucidate the relationship between sex-based biological differences between males and females and their impact on the development of effective antimalarial immunity in humans.

Of the variables we evaluated in addition to sex, baseline infection status and parasite density were most strongly associated with the rate of clearance of infection. Infections that were already present at the beginning of the study and persisted past the left censoring date may have been a non-random selection of well-established asymptomatic infections that were present at baseline at a higher frequency than average because they had a fundamentally different trajectory than newly established asymptomatic infections. Higher parasite densities were also associated with slower clearance of infection in both adjusted models, but the effect was most pronounced when the data were analyzed by infection event. This may be because low-density infections that we were unable to successfully genotype were only included in the infection event analysis, and these events tended to have short durations. The inclusion of parasite density in our multivariate models did not meaningfully alter associations between sex and duration of infection, providing evidence that the sex-based differences in duration were not mediated primarily by differences in parasite density in our cohort.

A limitation of our study is the statistical model's assumption that all infections clear at the same rate, which was necessary because we did not observe the beginning of most infections. To rely less on this assumption, we adjusted for baseline infections, allowing them to have a different rate of clearance than new infections; this did not change our primary finding of a difference in rate of clearance by sex. We also acknowledge there may be other unmeasured confounders, such as genetic hemoglobinopathies or sex-based differences in unreported outside antimalarial use, that could affect our results. Another caveat is that our findings may not be generalizable to areas with different transmission intensity given that our study was conducted in a very specific setting: an area with previously very high transmission intensity that has been greatly reduced in recent years by repeated rounds of IRS.

Additionally, because it is difficult to genotype low-density infections and because parasite densities fluctuate over time, we allowed several 'skips' in detection before declaring an infection cleared or the same clone in an individual a new infection with the same clone. This requires the assumption that re-infection with the same clone is relatively rare, which is reasonable given the high genetic diversity and low force of infection seen in this setting; this assumption has been made in other longitudinal genotyping studies (*Felger et al., 2012*; *Smith et al., 1999*). Our method may have biased us toward longer durations of infection and fewer new infections overall, but this is unlikely to have introduced any significant bias in the observed associations by sex. We performed a sensitivity analysis to ensure that changing the number of skips allowed did not significantly change the main finding regarding longer duration of infection in males and our findings were consistent. Given decreasing parasite densities over time and a setting with low force of infection and low EIR, assuming perfect recovery of clones is likely to artificially inflate the force of infection in this cohort. There were not enough infections to perform a rigorous analysis of the distribution of clones within and between households, but given that the overall force of infection was quite low, the probability of re-infection with the same clone already present in a participant from another member in the household (which could bias toward longer duration of infection) was low and unlikely to have introduced any significant bias by sex.

In summary, we estimated the clearance of asymptomatic *P. falciparum* infections by genotyping longitudinal samples from a cohort in Nagongera, Uganda, using sensitive amplicon deep-sequencing and found that females cleared their infections at a faster rate than males; this finding remained consistent when adjusting for age, baseline infection status, and parasite density. Furthermore, we found no conclusive evidence for a sex-based difference in exposure to infection, either behaviorally or by FOI. Though there have long been observed differences in malaria burden between the sexes, there is still little known about sex-based biological differences that may mediate immunity to malaria. Unfortunately, much reported malaria data is still sex-disaggregated. Our findings should encourage epidemiologists to better characterize sexual dimorphism in malaria and motivate increased research into biological explanations for sex-based differences in the host response to the malaria parasite.

## Materials and methods

### Study setting and population-level malaria control interventions

This cohort study was carried out in Nagongera sub-county, Tororo district, eastern Uganda, an area with historically high malaria transmission. However, 7 rounds of indoor residual spraying (IRS) from 2014 to 2019 have resulted in a significant decline in the burden of malaria (*Nankabirwa et al., 2019*). Pre-IRS, the daily human biting rate (HBR) was 34.3 and the annual entomological inoculation rate (EIR) was 238; after 5 years of IRS, in 2019 the daily HBR was 2.07 and overall annual EIR was 0.43 as reported by Nankabirwa et. al (*Nankabirwa et al., 2020*).

### Study design, enrollment, and follow-up

All members of 80 randomly selected households with at least two children were enrolled in October 2017 using a list generated by enumerating and mapping all households in Nagongera sub-county (*Nankabirwa et al., 2020*). The cohort was dynamic such that residents joining the household were enrolled and residents leaving the household were withdrawn. Data for this analysis was collected from October 1st, 2017 through March 31st, 2019; participants were included if they had at least 6 months of contiguous follow-up. Data from the 25 dynamically enrolled participants contributed to all analyses, including that of baseline infections. Participants were followed at a designated study clinic open daily from 8 AM to 5 PM. Participants were encouraged to seek all medical care at the study clinic and avoid the use of antimalarial medications outside of the study. Routine visits were conducted every 28 days and included a standardized clinical evaluation, assessment of overnight travel outside of Nagongera sub-county, and collection of blood by phlebotomy for detection of malaria parasites by microscopy and molecular studies. Participants came in for non-routine visits in the setting of illness. Blood smears were performed at enrollment, at all routine visits and at non-routine visits if the participant presented with fever or history of fever in the previous 24 hr. Participants with fever (>38.0°C tympanic) or history of fever in the previous 24 hr had a thick blood smear

read urgently. If the smear was positive, the patient was diagnosed with malaria and treated with artemether-lumefantrine. Participants with asymptomatic parasitemia as detected by qPCR or microscopy were not treated with antimalarials, consistent with Uganda national guidelines. Study participants were visited at home every 2 weeks to assess use of long-lasting insecticidal nets (LLINs) the previous night.

## Laboratory methods

Thick blood smears were stained with 2% Giemsa for 30 min and evaluated for the presence of asexual parasites and gametocytes. Parasite densities were calculated by counting the number of asexual parasites per 200 leukocytes (or per 500, if the count was less than 10 parasites per 200 leukocytes), assuming a leukocyte count of 8,000/µL. A thick blood smear was considered negative if examination of 100 high power fields revealed no asexual parasites. For quality control, all slides were read by a second microscopist, and a third reviewer settled any discrepant readings. In our experienced microscopists' hands, the lower limit of detection is approximately 20–50 parasites/µL.

For qPCR and genotyping, we collected 200 µL of blood at enrollment, at each routine visit, and at the time of malaria diagnosis. DNA was extracted using the PureLink Genomic DNA Mini Kit (Invitrogen) and parasitemia was quantified using an ultrasensitive varATS qPCR assay with a lower limit of detection of 0.05 parasites/µL (*Hofmann et al., 2015*). Samples with a parasite density >= 0.1 parasites/µL blood were genotyped via amplicon deep-sequencing. All samples positive for asexual parasites by microscopy but negative for *P. falciparum* by qPCR were tested for the presence of non-falciparum species using nested PCR (*Snounou et al., 1993*).

### Sequencing library preparation

Hemi-nested PCR was used to amplify a 236 base-pair segment of apical membrane antigen 1 (AMA-1) using a published protocol (*Miller et al., 2017*), with modifications (*Supplementary file 1a*). Samples were amplified in duplicate, indexed, pooled, and purified by bead cleaning. Sequencing was performed on an Illumina MiSeq platform (250 bp paired-end).

## Bioinformatics methods

Data extraction, processing, and haplotype clustering were performed using SeekDeep (*Hathaway et al., 2018*), followed by additional filtering (*Lerch et al., 2019*). *Supplementary file 1b* shows the full bioinformatics workflow.

## Data analysis

A clone was defined as a genetically identical group of parasites, for example with identical haplotypes. Because polyclonal infections can occur due to co-infection (one mosquito bite transmitting multiple clones) or superinfection (multiple bites), we analyzed the infection data both by clone and by infection event. For analysis by clone, each unique clone was counted as an infection and each clone's disappearance as a clearance event. For analysis by infection event, any new clones seen within 3 visits of the date of the first newly detected clone(s) were grouped together and considered one new 'infection event.' Clearance of infection for these events required that all clones in the group be absent. A baseline infection was defined as a clone or infection event (group of clones) detected in the first 60 days of observation of a participant. New infections were defined as a new clone or infection event (group of clones) detected in a participant after day 60.

Imperfect detection of *P. falciparum* clones is known to be a limitation of all PCR-based genotyping methods; both biological factors, such as deep organ sequestration of clones in the lifecycle of the parasite, and methodological factors, such as difficulty amplifying minority clones (especially at lower parasite densities), results in imperfect detectability of all clones present in a single blood sample drawn on a single day (*Felger et al., 2012*; *Koepfli and Mueller, 2017*; *Nguyen et al., 2018*; *Koepfli et al., 2011*; *Ross et al., 2012*). Though amplicon deep-sequencing is less likely than other methods to miss minority clones (*Lerch et al., 2017*), detectability was a particular concern in our study, in which the majority of infections were submicroscopic and parasite densities declined over time [*Supplementary file 1c*]. To account for the fact that a clone might be missed in a single sample due to fluctuations in parasite density and/or methodological limitations, we allowed 3 'skips' in detection and classified an infection as cleared only if it was not identified in four contiguous

samples from routine visits. If the infection (as determined by clone or by infection event) was absent for four routine samples in a row, the last date of detection would be the end date of that infection. Therefore, for a clone that had previously infected an individual to be classified as a new infection, it had to have been absent from that individual for at least four routine visits. Additional details are found in *Supplementary file 1d*. This decision was supported by the fact that the diversity of the genotyped AMA-1 amplicon was quite high (expected heterozygosity = 0.949), resulting in a low probability (5.1%) for infection with the same clone by chance.

Data analysis was conducted in R (*R Development Core Team, 2019*) and Python (*Python Language Reference, 2020*). Travel was self-reported at routine visits, and LLIN use on the previous night was reported by field entomology teams who surveyed households every two weeks. Microscopic parasite prevalence was defined by the number of smear-positive routine visits over all routine visits. Parasite prevalence by qPCR was defined as the number of qPCR-positive visits over all routine visits. Thus, these measures represent the average prevalence during routine visits. Prevalence ratios were computed using Poisson regression with generalized estimating equations to adjust for repeated measures. Comparison of parasite density by sex was made using linear regression with generalized estimating equations to adjust for repeated measures. Force of infection (FOI) was defined as the number of new infections, including malaria episodes, divided by person time. Since the start of baseline infections was not observed, baseline infections were not included in calculating FOI. Poisson regression with generalized estimating equations to account for repeated measures was used to estimate malaria incidence, FOI, and to calculate incident rate ratios (IRR) for malaria and FOI. Hazards for clearance of untreated, asymptomatic infections were estimated using time-to-event models (shared frailty models fit using R package 'frailtypack,' version 2.12.2) (*Rondeau and Gonzalez, 2005*; *Rondeau et al., 2012*). Infections were censored for this analysis if they were only observed in the first three months or the last three months (before January 01, 2018 and after January 01, 2019) because they were not observed for long enough to determine whether clearance occurred. If an infection was observed for only one timepoint, it was assigned a duration of 14 days. These models assumed a constant hazard of clearance and included random effects to account for repeated measures in individuals. Parasite density was included in the model as a time-varying covariate. Duration of infection in days was calculated as 1/adjusted hazard.

## Data accessibility

Data from the cohort study is available through an open-access clinical epidemiology database resource, ClinEpiDB at https://clinepidb.org/ce/app/record/dataset/DS_51b40fe2e2. Genotyping data and code used to generate tables and figures is available on GitHub (*Briggs, 2020*; copy archived at swh:1:rev:cf6c3256e609f4f136fc8d90f9cae1d61d6d8d63).

## Additional information

### Competing interests

Isabel Rodriguez-Barraquer: Reviewing editor, *eLife*. The other authors declare that no competing interests exist.

### Funding

| Funder | Grant reference number | Author |
|--------|------------------------|--------|
| National Institute of Allergy and Infectious Diseases | T32 AI007641-16 | Jessica Briggs |
| National Institute of Allergy and Infectious Diseases | U19AI089674 | Grant Dorsey |
| Fogarty International Center | K43TW010365 | Joaniter I Nankabirwa |
| Fogarty International Center | D43TW010526 | Emmanuel Arinaitwe |

The funders had no role in study design, data collection and interpretation, or the decision to submit the work for publication.

## Author contributions
Jessica Briggs, Conceptualization, Data curation, Formal analysis, Methodology, Writing - original draft, Writing - review and editing; Noam Teyssier, Formal analysis, Visualization, Methodology; Joaniter I Nankabirwa, Resources, Supervision, Project administration; John Rek, Emmanuel Arinaitwe, Supervision, Project administration; Prasanna Jagannathan, Conceptualization, Supervision, Methodology, Writing - review and editing; Teun Bousema, Phillip J Rosenthal, Supervision, Funding acquisition, Writing - review and editing; Chris Drakeley, Resources, Supervision, Funding acquisition, Writing - review and editing; Margaret Murray, David Smith, Conceptualization, Writing - review and editing; Emily Crawford, Resources, Supervision; Nicholas Hathaway, Software, Methodology, Writing - review and editing; Sarah G Staedke, Resources, Supervision, Writing - review and editing; Moses Kamya, Resources, Supervision, Funding acquisition, Project administration, Writing - review and editing; Grant Dorsey, Conceptualization, Resources, Supervision, Funding acquisition, Project administration, Writing - review and editing; Isabel Rodriguez-Barraquer, Conceptualization, Formal analysis, Supervision, Validation, Methodology, Writing - review and editing; Bryan Greenhouse, Conceptualization, Formal analysis, Supervision, Funding acquisition, Methodology, Writing - original draft, Project administration, Writing - review and editing

## Author ORCIDs
Jessica Briggs (iD) https://orcid.org/0000-0002-8078-3898
Prasanna Jagannathan (iD) http://orcid.org/0000-0001-6305-758X
Teun Bousema (iD) http://orcid.org/0000-0003-2666-094X
Chris Drakeley (iD) http://orcid.org/0000-0003-4863-075X
Isabel Rodriguez-Barraquer (iD) https://orcid.org/0000-0001-6784-1021

## Ethics
Human subjects: IRB approval for the PRISM2 cohort study was obtained in Uganda (UNCST HS-119ES, SOMREC: 2017-099), UK (LSHTM: 14266), and US (UCSF IRB: 17-22544, Stanford: UCSF IRB reliance). Informed consent was obtained from all participants prior to enrollment in the study per IRB guidelines.

## Decision letter and Author response
Decision letter https://doi.org/10.7554/eLife.59872.sa1
Author response https://doi.org/10.7554/eLife.59872.sa2

# Additional files

## Supplementary files
• Supplementary file 1. (**a**) AMA-1 hemi-nested PCR protocol for amplicon deep-sequencing. (**b**) Bioinformatics workflow. (**c**) Declining qPCR density over time in the cohort. (**d**) Detailed explanation of skip rule criteria. (**e**) Haplotype sequences and frequencies. (**f**) Sensitivity analysis of molecular force of infection: *Table 2* replicated using 2 skips or 1 skip. (**g**) Sensitivity analysis of duration of infection: *Table 3* replicated using 2 skips or 1 skip.

• Transparent reporting form

## Data availability
Data from the PRISM2 cohort study is available through a novel open-access clinical epidemiology database resource here: https://clinepidb.org/ce/app/record/dataset/DS_51b40fe2e2. Sequencing data is available on Github at https://github.com/EPPIcenter/sex_based_differences as referenced in the paper. In addition. all sequences of haplotypes are included in Supplementary file 1.

The following dataset was generated:

| Author(s) | Year | Dataset title | Dataset URL | Database and Identifier |
|---|---|---|---|---|
| Dorsey G | 2020 | PRISM2 ICEMR Cohort | https://clinepidb.org/ce/app/record/dataset/DS_51b40fe2e2 | ClinEpiDB, DS_51b40fe2e2 |

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
