## [Decision Letter]

**Acceptance summary:**

Studies on sex-based differences in immune responses, in general, have established that females show enhanced activity in comparison to males. Recently, several studies have confirmed a male bias in the prevalence of human parasitic infections. The current study explores the possibility of sex dimorphism in susceptibility to Plasmodium malaria. On the basis of results from longitudinal studies conducted in Uganda, females cleared their malaria infections at a faster rate than males. These observations provide an impetus for increased research into biological explanations for sex-based differences in the host response to the malaria parasite.

**Decision letter after peer review:**

Thank you for submitting your article "Sex-based differences in clearance of chronic *Plasmodium falciparum* infection" for consideration by *eLife*. Your article has been reviewed by three peer reviewers, and the evaluation has been overseen by a Reviewing Editor and a Senior Editor. The following individual involved in review of your submission has agreed to reveal their identity: Richard Price.

As is customary in *eLife*, the reviewers have discussed their critiques with one another. What follows below is the Reviewing Editor's edited compilation of the essential and ancillary points provided by reviewers in their critiques and in their interaction post-review. Please submit a revised version that addresses these concerns directly. Although we expect that you will address these comments in your response letter we also need to see the corresponding revision in the text of the manuscript. Some of the reviewers' comments may seem to be simple queries or challenges that do not prompt revisions to the text. Please keep in mind, however, that readers may have the same perspective as the reviewers. Therefore, it is essential that you attempt to amend or expand the text to clarify the narrative accordingly.

As the editors have judged that your manuscript is of interest, but as described below additional work is required before it is published, we would like to draw your attention to changes in our revision policy that we have made in response to COVID-19 (https://elifesciences.org/articles/57162). First, because many researchers have temporarily lost access to the labs, we will give authors as much time as they need to submit revised manuscripts. We are also offering, if you choose, to post the manuscript to bioRxiv (if it is not already there) along with this decision letter and a formal designation that the manuscript is "in revision at *eLife*". Please let us know if you would like to pursue this option. (If your work is more suitable for medRxiv, you will need to post the preprint yourself, as the mechanisms for us to do so are still in development.)

Summary:

The reviewers considered these findings suggesting a bias in *Plasmodium falciparum* prevalence in males versus females to be very interesting and potentially important to a wide scientific audience. As the authors conclude, the sexual differences are caused by higher prevalence of infection in males because of longer infection carriage or slower clearance of infection. The convincing demonstration of true sexual dimorphism (as opposed to gender-related behavioral/occupational differences) in malaria risk would be an important scientific advance. Despite strengths of the study, the reviewers expressed several critical concerns which include lack of data, robustness of statistical analysis and some basic methodology. These important issues preclude formation of strong conclusions that males and females differed in malaria risk for intrinsic biological reasons versus external factors.

Essential revisions:

A general question for this study is the scope and generalizability of the problem the authors are addressing; while many earlier studies reported malaria epidemiology, only a handful reported sexual dimorphism in risk. The papers cited by the authors describe different (and inconsistent) evidence for sexual dimorphism. Pathak et al., 2012, cited in the paper states: "Sexual dimorphism does not exist in hyperendemic regions for both *Plasmodium falciparum* and *P. vivax* infections, although some reports note an increased parasite density in pubertal and post-pubertal males".

Issues with the age categorization: why were these age bands chosen? If hormonal differences play a role in gender differences then surely this would increase after puberty. Perhaps it did and that is why higher difference is emerging in the 5-15 group. The three categories lose important data on the age sex distributions that could be highly relevant to the analysis. Surely 5-12yrs would have helped explore the puberty issue. Another approach would be to retain age as a continuous variable. Either way this doesn't explain the smaller difference in adults – although there were low numbers of males and few events. Did the age limit extend into older/menopausal women and might this have contributed to the findings? Please discuss these observations in the conclusions. The discussion also needs to acknowledge the limitations of the study and how assumptions and residual confounding may have impacted upon the findings.

Swimmers plots or similar figure may be helpful to display duration of individual infections over time. These could be stratified by sex, would show the number and durations of "infections" and of "clones", and individuals could be sorted by age so that the relevant age groups could be indicated.

Please include information if baseline infections differed by sex. Baseline infections have significantly longer duration than new infections (which is counter-intuitive to this reviewer) and constituted a majority of the infections in their analyses. Did the authors see similar patterns of sexual dimorphism in their baseline infections as well as their new infections?

Data on important confounders are missing, for example, genetics/ hemoglobinopathies, drug use (by report and by drug levels during "asymptomatic" infections), time since residual spraying if done.

Terms used in the paper have overlapping meanings ("clone", "infection", "parasitemia", "malaria infection", "malaria", "asymptomatic malaria", "symptomatic malaria", etc) and not all are clearly defined-would be good to define each and use consistently. The definitions for "clearance" require some clarification.

Other major concerns surround the robustness of the conclusions based on the statistical analysis:

1) No direct statistical test is given to show that prevalence differs between the sexes after adjusting for behavioural factors (and age etc).

2) The analysis of infection clearance and duration of infection (and even the FOI) are all based on the arbitrary assumption that 4 negative routine visits are needed before an infection is deemed to have been cleared. How many of the results change when this assumption is relaxed?

3) The analysis of FOI between sexes may be confounded by what is classed as a "new infection". Does this result still hold when the definition of new infection is altered?

These are suggestions for validating the robustness of the conclusions.

1) The authors do not directly test whether sex is still a predictor of higher prevalence after other factors are considered, which one would expect to see in the typical analysis in order to make the claim that behavioural factors do not explain the difference between men and women. Instead, the authors argue by looking at a table of the mean characteristics of the population what is and is not different between men and women (Table 1). Their conclusion on the basis of a lack of significance between behavioural pattern beteween sexes this is not a statistical argument. If the authors wish to show that, after accounting for behavioural differences, there is still a link between sex and prevalence, they need a full statistical model that includes all factors. The authors should run a logistic regression looking at the predictors of infection status in individuals, and showing that sex is a significant predictor. Covariates in this model should be sex, age, the behavioural data (e.g. bednet usage, travel, etc), time point. The model should also include random-effects to account for repeated measures within individuals. If no other factors are included, it could be assumed that sex would be a predictor of infection. However, forwards and/or backwards regression could then be used to ask the question "is sex a significant predictor of infection once adjusting for other behavioural factors are included in the model?" Similar analysis is performed on the clearance data so this should be easily achievable for the authors. If sex is not a predictor of infection after adjusting for age and behavioural factors, then one might conclude that risk of detecting an infection is well explained by behavioural factors and not sex differences. If sex is still required to explain the data then that would be evidence for a difference in prevalence that is sex based on not behavioural.

2) The authors argue that women have faster clearance times of asymptotic infection but this analysis is directly dependent on the definition of when an infection is cleared. Follow-up is every 28 days in this study and the end of an infection was defined as a clonotype being absent for 4 routine visits (i.e. someone might have a clone present at one visit, and at a visit 112 days later, but not detected at all in-between and it would be considered the same infection). If one is reinfected with the same clone (which might be very probably in an environment where 10 of the clones detected account for 55% of all observed clones) it is possible to count a new infection as one long infection. This issue may both artificially prolonging clearance times in the analysis and reduce the number of "new infections" (i.e. lower the FOI calculated) and may bias observations for men and women. This demand for 4 consecutive negative visits before an infection is deemed to have been cleared is an arbitrary choice and to be believable the conclusions of the analysis should not depend strongly on this assumption. For this reason, the authors must check that varying this criteria doesn't alter the results too strongly.

a) The authors should explore how this definition of clearance (being 4 negative routine visits) impacts the conclusions. What if only one negative time point was used as the definition of the end of an infection? How would this affect the FOI for men and women, and how would it impact the clearance times observed? Since it is not possible to really know when an infection ended, and a new infection began, it is important to test that the conclusions are robust to these assumptions.

b) It may be that behaviour patterns impact the likelihood of being infected with the same clonotype twice. For example, if one group does more travel than another group, then they may be more likely to be infected with different clonotypes. This diversity of clonotypes would make an individual look like they had shorter durations of infection compared with a group that travels less and is re-infected over and over with the same clonotype. i.e. Women who travelled more may only look like they have shorter infection because they are not re-infected with the same clone as much as the men who travelled less. To test this, the authors should look at the distribution of clonotype frequencies and diversity between men and women (e.g. do women tend to have infection with a greater diversity in clonotypes than men?). This might also make men look like they have a lower FOI than they really do, and women a higher FOI than they really do, and it may make it seem like the two groups have the same FOI when actually they are very different.

c) The issue of 4 negatives being required for "clearance" may explain why the infection present in individuals at enrollment is longer than "new infection". This result would change if the end of an infection was defined as the first negative rather than requiring 4 negatives. This may indicate a bias in the analysis towards counting things as one infection that are actually reinfection with the same clonotype.

3) The authors should confirm that there is no difference in FOI between the sexes. Even though not significant, there appears to be a trend towards higher FOI in men compared with women in all three age groups (Table 2). This is important because it is not clear how different the FOI in men and women would need to be in order for it to manifest in the differences in prevalence that exist between men and women. It may be that there is really a difference in prevalence observed and that this study does not have statistical power to detect such a difference. Also, calculating FOI as the authors have done requires knowing "new infections" as opposed to just the carriage of old infection, and the definition of what is a new infection is going to influence these conclusions. The authors should further validate this conclusion from their data by: (a) changing their definition of new infection (as in point 2) and recalculating the FOI and determining if there is any significant difference between the sexes. (b) performing a time-to-next-infection analysis (survival analysis, e.g. cox regression), rather than calculate the FOI as new infections over person time. The latter would involve looking at the time until the next new clone appears in each person in a survival model (with appropriate censoring). The question would then be "is sex a significant covariate in a survival model of time to next infection (after including other behavioural factors as well, such as travel)?". If these two other approaches also reveal no difference in FOI between the sexes, then the conclusion that there is no difference in infection pressure on the sexes would be more robust/convincing.

[Editors' note: further revisions were suggested prior to acceptance, as described below.]

Thank you for resubmitting your work entitled "Sex-based differences in clearance of chronic *Plasmodium falciparum* infection" for further consideration by *eLife*. Your revised article has been evaluated by a Senior Editor and a Reviewing Editor.

The manuscript has been improved but there are some remaining issues that need to be addressed before acceptance, as outlined below:

1) The information on baseline infections, a potential confounder to these analyses, still needs further clarification.

Subsection “Cohort participants and *P. falciparum* infections”: "25 participants.… were enrolled after initial enrollment…" Please state whether their baseline data contributed to analyses of baseline infections. It seems so from subsection “Data analysis” but should be stated explicitly.

Subsection “Cohort participants and *P. falciparum* infections” paragraph two: "35 samples had very low density infections (< 1 parasite/μL) that could not be genotyped and had infections characterized at the event level only". Did these come from 35 unique individuals who had a single infection? And do these represent the difference between 149 individuals with infections versus 114 individuals who had 822 typable samples? If so, please state this explicitly. It appears so from Figure 1.

"117/185.… baseline infections in males". Please state the proportion of all infections included in the analysis that were baseline infections, for males and females separately, at the clone and the infection event level. Baseline infections are an important confounder in this study because they differ from other infections and disproportionately occur in males at the clone level.

Subsection “Behavioral malaria risk factors and measures of malaria burden”: "differences in prevalence…" Prevalence is defined in subsection “Data analysis”, but it might be helpful to use the term "period prevalence" or a simple explanation here, since many non-epidemiology readers may assume that prevalence here refers to a point prevalence.

In the same section please define COI.

Subsection “Force of infection by age and sex”: "Force of infection…" Did FOI include baseline infections? Not clear from the definition in subsection “Data analysis”. Please state clearly at the first instance of the term. I assume baseline infections are not included in FOI since it is not possible to know when these started.

Subsection “Rate of clearance of infection and duration of infection by sex”: "68/105 (64.8%) baseline infections in males…"- as above, please state the proportion of all infections included in the analysis that were baseline infections, for males and females separately.

Discussion: "stronger effect in adults than in school-aged children, which was not seen in this cohort". Studies in adolescents have shown a strong relationship between control of parasitemia and adrenarche/DHEAS levels. The authors fail to cite the existing literature showing adrenarche and increasing DHEAS levels which correspond to malaria resistance in males and females. Citing these studies will contribute to the findings here, inasmuch as adrenarche starts earlier in females than males, and the relationship of DHEAS to malaria resistance carries over an extended age window.

Discussion: "a unique area with previously very high transmission intensity that has been greatly reduced in recent years by repeated rounds of IRS." This is not unique. The Garki project provides a similar example. In the Garki project report (p 155), there is no prominent difference in males vs females before the intervention; further, there is a prominent difference after the intervention in the intervention communities, but not in the control communities. This seems relevant to the current study and should be discussed. The Garki project is specifically called out in paragraph three of the Discussion.

Subsection “Data analysis”: "Infections were censored…" Censored for the clearance analysis only, or censored for both FOI and clearance analysis? Please state clearly.

Table 1: "Episodes of malaria**, (incidence*)" should just be "Episodes of malaria**", correct?

2) The authors point out that there is a low likelihood of reinfection with the same genotype. Addressing the issue of clonotype distribution within and between households may not be possible to include easily in this study. For that reason, please consider adding a note of this limitation to the discussion, since your analysis of the duration of infections (which relies on time infected with the same clonotype) may be impacted if there were a strong tendency of individuals to be reinfected with the same clones.

3) Please consider adding the following:

a) The literature appears conflicting, with significant confounding factors (such as treatment seeking) and study design. There may also be a species difference. See Tjitra et al. PlosMed 2008 Figure 2.

b) Inherent genetic factors remain a plausible explanation and X-linked disorders would warrant particular scrutiny. One point of correct for G6PD, is that although males are at higher risk of severe deficiency (hemizygotes ~5%), the greatest proportion are

actually heterozygous females with intermediate deficiency (15-20%). If intermediate deficiency is sufficient to effect clinical susceptibility to disease then this could be an interesting sub group.

---

## [Author Response]

Essential revisions:A general question for this study is the scope and generalizability of the problem the authors are addressing; while many earlier studies reported malaria epidemiology, only a handful reported sexual dimorphism in risk. The papers cited by the authors describe different (and inconsistent) evidence for sexual dimorphism. Pathak et al., 2012 cited in the paper states: "Sexual dimorphism does not exist in hyperendemic regions for both *Plasmodium falciparum* and *P. vivax* infections, although some reports note an increased parasite density in pubertal and post-pubertal males".

We have expanded the Introduction to better review the evidence. Although sex-specific differences in malaria have not been uniformly described in younger children, the preponderance of the evidence over all regions is in favor of a male bias in malaria infections in school-aged children and adults (excepting pregnant women). The comprehensive Garki study (Molineaux et al., 1980) found that after 5 years of age, males have higher average parasite prevalence than females; several of the differences were statistically significant, including a greater than 2-fold higher prevalence of *P. falciparum* in the age-groups 9-18. Similarly, studies in Ghana (Landgraf et al., 1994), Sudan (Creasy 2004), and India (Pathak et al., 2012) have reported higher test positivity rates in school-aged children when comparing males with females. In addition, the majority of studies on malaria incidence and/or prevalence that stratify by sex in adulthood found a male bias in the observed measure of disease (Molineaux et al., 1980, Meek, 1988; Abdalla, Malik, and Ali, 2007; Pathak et al., 2012). Other studies have found a male bias in malaria incidence and/or prevalence but do not explicitly stratify by age group (Carmago 1996, Moon and ho, 2001; Mulu et al., 2013). Less commonly, researchers have found no sex-specific difference in adulthood in burden of malaria (Mendis et al., 1990, Sur 2006). Unfortunately, many papers do not stratify malaria prevalence or incidence by sex as they primarily focus on age stratification. In addition, the data that are published on malaria prevalence are often based on microscopy data alone. For example, in our data, Table 1 does not show a statistically significant difference between microscopy prevalence by sex using a simple comparison of proportions but does show a significant difference by qPCR prevalence, which is a much more sensitive diagnostic. If in fact higher malaria prevalence in males is driven by longer infections, then the tail end of these infections would most likely be lower density and more difficult to detect without molecular diagnostics.

Issues with the age categorization: why were these age bands chosen? If hormonal differences play a role in gender differences then surely this would increase after puberty. Perhaps it did and that is why higher difference is emerging in the 5-15 group. The three categories lose important data on the age sex distributions that could be highly relevant to the analysis. Surely 5-12yrs would have helped explore the puberty issue. Another approach would be to retain age as a continuous variable. Either way this doesn't explain the smaller difference in adults – although there were low numbers of males and few events. Did the age limit extend into older/menopausal women and might this have contributed to the findings? Please discuss these observations in the conclusions. The discussion also needs to acknowledge the limitations of the study and how assumptions and residual confounding may have impacted upon the findings.Swimmers plots or similar figure may be helpful to display duration of individual infections over time. These could be stratified by sex, would show the number and durations of "infections" and of "clones", and individuals could be sorted by age so that the relevant age groups could be indicated.

First, we apologize that the last 2 paragraphs of the manuscript were somehow lost when formatting for submission (this included the limitations paragraph, which we have now expanded to two paragraphs based on reviewers’ comments).

Age groups were initially chosen to be consistent with numerous other publications we and others have published on this and similar cohorts, and to reflect salient differences in the overall epidemiology of disease between young children, school-age children, and adults. When changing the adjusted model so that age in continuous years is used instead of age categories, there is no meaningful change to the finding of faster clearance of infection in females (HR = 1.89 by clone and HR = 1.96 by infection event); however this does not answer the question of interaction by age. Based on the reviewers’ comments, we have also performed the analysis with age bands <8 years of age, 8-13 years, and 13+ years given that the onset of puberty occurs on average at approximately age 8-9 in females and age 9-10 in males in the US and may occur slightly later in Uganda. Changing the age bands did not meaningfully alter our findings (adjusted hazard ratio (HR) for clearance of infection by clone in females vs males= 1.64, adjusted HR by infection event = 2.13). To look for evidence of interaction by age, we stratified by these modified age bands and calculated unadjusted HR for clearance of infection in females versus males. By clone, unadjusted hazard ratio (HR) for clearance of infection in females versus males was 1.69 (0.45 – 6.34) in children <8, 2.01 (0.81 – 5.00) in children 8-13, and 1.58 (0.95 – 2.63) in those over the age of 13. Performing the same unadjusted stratified analysis for the age bands used in our paper gave HR for clearance of clones in females vs males of 2.31 (0.73-7.28) in children <5, 1.92 (0.97 – 3.78) in children 5-15, and 1.42 (0.70 – 2.87) in those 16 and older. By infection event, HR for clearance of infection using modified age bands was 0.79 (0.23 – 2.73) in children <8, 1.56 (0.19 – 12.49) in children 8-13, and 3.54 (1.68 – 7.48) in those over 13. Performing the same unadjusted stratified analysis for the age bands used in our paper, HR for clearance of infection events in females was 1.09 (0.31-3.78) in children < 5, 1.50 (0.42 – 5.36) in children 5-15, and 3.79 (1.48. – 9.69) in those 16 and older. Regardless of whether the original age bands or the modified age bands are used, similar trends are seen. The trend toward more evidence of sexual dimorphism in the highest age group by infection event but not by clone is likely explained by the fact that adults are known to have infections with significantly lower parasite densities than children; while we can detect infections down to 0.05 parasites/microliter using ultrasensitive qPCR, we are only able to genotype infections at >=0.1 parasite/microliter. While there is no definitive evidence for an interaction between age and sex when it is included in the final adjusted model, our sample size limits the power that we have to detect this type of interaction. Finally, as pointed out by the reviewers, post-menopausal women were included in the highest age band which could confound the findings in that age category, but are not of sufficient numbers to make any meaningful conclusions in subgroup analysis. We added these points to the Discussion.

Please include information if baseline infections differed by sex. Baseline infections have significantly longer duration than new infections (which is counter-intuitive to this reviewer) and constituted a majority of the infections in their analyses. Did the authors see similar patterns of sexual dimorphism in their baseline infections as well as their new infections?

Information on baseline infections by sex has now been added to the manuscript in two places in the Results. Males had more baseline infections by clone but not by infection event, which indicates higher complexity of infection in their baseline infections, consistent with our finding of longer duration of infection.

Data on important confounders are missing, for example, genetics/ hemoglobinopathies, drug use (by report and by drug levels during "asymptomatic" infections), time since residual spraying if done.

We do not have data available on genetic hemoglobinopathies that affect susceptibility to malaria for this cohort. G6PD deficiency, which is an X-linked recessive disorder, is more common in males. There is no reason to suspect a sex bias in sickle-cell anemia (or trait) or α thalassemia since they are not sex-linked traits; although this could be an empirical confounder if there was an unequal distribution by random chance in this cohort. We have added this to the study limitations as mentioned in the Discussion. Likewise, we do not have data on drug levels in this cohort. However, our clinic is open 7 days a week, free, high-quality care is provided, and transport is reimbursed, giving participants a little incentive for seeking outside care. We have performed studies of this design for many years and receipt of outside care has never been a major issue. We do ask all participants at every routine visit if they have taken any outside antimalarials. Only for 4 participants was this ever reported (only once per participant). I have added a line acknowledging this in the Results. All IRS in this district was performed over 1 month and so there was little difference between households in time since residual spraying.

Terms used in the paper have overlapping meanings ("clone", "infection", "parasitemia", "malaria infection", "malaria", "asymptomatic malaria", "symptomatic malaria", etc) and not all are clearly defined-would be good to define each and use consistently. The definitions for "clearance" require some clarification.

We apologize for any confusion in our terminology. An infection with malaria parasites (measured as detectable parasitemia) can be caused by a single strain (which we refer to as clone) or by multiple clones. Analyses can be performed at the clone level or at the infection-event level (which groups various clones into a single infection); therefore it is necessary to use the word “infection” at a more general level at times. I have made clarifying edits regarding clone/infection throughout the paper in the Materials and methods and Results. “Asymptomatic malaria” is not a term used in our paper. Episodes of clinical malaria are clearly defined in the Materials and methods and in Table 1. Parasitemia is used to denote the presence of malaria parasites in the blood and is used only in the context of prevalence of infection (Table 1 and first paragraph of Results).

Other major concerns surround the robustness of the conclusions based on the statistical analysis:1) No direct statistical test is given to show that prevalence differs between the sexes after adjusting for behavioural factors (and age etc).

Using generalized estimating equations to account for clustering by individual, prevalence ratio (PR) of *P. falciparum* parasitemia by microscopy in females versus males across all age categories was 0.49 (95% CI 0.26-0.90, p = 0.02), with relative differences in prevalence most pronounced in the oldest age group. Similar findings were seen when prevalence was assessed by ultrasensitive qPCR, with PR = 0.64 in females vs. males (95% CI 0.43-0.96, p = 0.03), again with the largest differences seen in the oldest age group. Including age category, LLIN use, and travel in a Poisson regression model using generalized estimating equations to account for clustering by individual did not qualitatively change the parasite prevalence ratio in females versus males, for microscopic parasitemia (PR in females vs. males = 0.57, 95% CI 0.42-0.77, p <0.001) or for submicroscopic parasitemia (PR in females vs. males = 0.67, 95% CI 0.60-0.76, p <0.001). This is addressed now in the text. In fact, including behavioral factors in the model increased QIC (decreased model fit), with the majority of variance in the model explained by age category and sex alone.

2) The analysis of infection clearance and duration of infection (and even the FOI) are all based on the arbitrary assumption that 4 negative routine visits are needed before an infection is deemed to have been cleared. How many of the results change when this assumption is relaxed?

We thank the reviewers for this raising this important point, as this is something we have spent a lot of time discussing and trying to figure out. While unfortunately the sample size of this study (and the number of observed infections) limits our capacity to model the clearance and re-infection processes jointly, the assumption allowing a maximum of 3 skips to be considered the same infection was not entirely arbitrary, as now discussed in the text. The decision was motivated by competing probabilities of having not detected a persistent infection at certain timepoints (low detectability) versus a new infection occurring, considering that: 1) diversity of the genotyped locus was quite high, resulting in a low probability (5.1%) for the different parasites having the same genotype by chance; 2) force of infection was low in the study, making it less likely that infection by an identical clone was new vs persistent; and 3) as a result parasite density was low in these long duration, “old” infections, making it more probable that persistent infections present would drop below the limit of detection and be missed at multiple timepoints. The reviewer’s suggestion to perform a sensitivity analysis with a fewer number of skips is a good suggestion. We have now performed this, and the main results do not change (i.e., there is still a faster clearance of infection in females) when the data are analyzed allowing for 2 skips or 1 skip. These data are now referred to in the manuscript and available in the Supplementary file 1. Notably, multiple prior longitudinal genotyping studies have made the same assumption that re-infection with the same clone is relatively rare, which is reasonable given the high genetic diversity seen in this setting and low force of infection; these are referred to in the Discussion.

3) The analysis of FOI between sexes may be confounded by what is classed as a "new infection". Does this result still hold when the definition of new infection is altered?

Fundamentally, when you change the number of skips allowed, some of those clones that were previously considered part of a long infection will become classified as new infections (likely falsely, given the considerations above). Thus, FOI increases for all groups. Tables showing FOI for 2 skips and 1 skip are provided in Supplementary file 1. There remains a trend toward higher FOI in males when Table 2 is replicated with 2 skips or 1 skip, but even when performing the analysis with 1 skip, there is no significant difference in FOI between the sexes (IRR for females vs. males when adjusted for age category = 0.71, 95% CI 0.41 to 1.24 by clone and IRR = 0.75, 95% CI 0.47 to 1.22 by infection event). There may be a difference by sex in the force of infection that we are unable to detect due to small number of new infections (overall low FOI). Notably, no matter how many skips are chosen, there remains sexual dimorphism in duration of infection.

Also, we would like to note a small correction to Table 2; we had mistakenly not accounted for repeated measures in our original estimation of FOI. The results have changed very slightly on account of this, and the Materials and methods have been updated to reflect the correct technique used (Poisson regression with generalized estimating equations).

These are suggestions for validating the robustness of the conclusions.1) The authors do not directly test whether sex is still a predictor of higher prevalence after other factors are considered, which one would expect to see in the typical analysis in order to make the claim that behavioural factors do not explain the difference between men and women. Instead, the authors argue by looking at a table of the mean characteristics of the population what is and is not different between men and women (Table 1). Their conclusion on the basis of a lack of significance between behavioural pattern beteween sexes this is not a statistical argument. If the authors wish to show that, after accounting for behavioural differences, there is still a link between sex and prevalence, they need a full statistical model that includes all factors. The authors should run a logistic regression looking at the predictors of infection status in individuals, and showing that sex is a significant predictor. Covariates in this model should be sex, age, the behavioural data (e.g. bednet usage, travel, etc), time point. The model should also include random-effects to account for repeated measures within individuals. If no other factors are included, it could be assumed that sex would be a predictor of infection. However, forwards and/or backwards regression could then be used to ask the question "is sex a significant predictor of infection once adjusting for other behavioural factors are included in the model?" Similar analysis is performed on the clearance data so this should be easily achievable for the authors. If sex is not a predictor of infection after adjusting for age and behavioural factors, then one might conclude that risk of detecting an infection is well explained by behavioural factors and not sex differences. If sex is still required to explain the data then that would be evidence for a difference in prevalence that is sex based on not behavioural.

This is addressed above under response #1.

2) The authors argue that women have faster clearance times of asymptotic infection but this analysis is directly dependent on the definition of when an infection is cleared. Follow-up is every 28 days in this study and the end of an infection was defined as a clonotype being absent for 4 routine visits (i.e. someone might have a clone present at one visit, and at a visit 112 days later, but not detected at all in-between and it would be considered the same infection). If one is reinfected with the same clone (which might be very probably in an environment where 10 of the clones detected account for 55% of all observed clones) it is possible to count a new infection as one long infection. This issue may both artificially prolonging clearance times in the analysis and reduce the number of "new infections" (i.e. lower the FOI calculated) and may bias observations for men and women. This demand for 4 consecutive negative visits before an infection is deemed to have been cleared is an arbitrary choice and to be believable the conclusions of the analysis should not depend strongly on this assumption. For this reason, the authors must check that varying this criteria doesn't alter the results too strongly.

This has been addressed above.

a) The authors should explore how this definition of clearance (being 4 negative routine visits) impacts the conclusions. What if only one negative time point was used as the definition of the end of an infection? How would this affect the FOI for men and women, and how would it impact the clearance times observed? Since it is not possible to really know when an infection ended, and a new infection began, it is important to test that the conclusions are robust to these assumptions.

These points are addressed above.

b) It may be that behaviour patterns impact the likelihood of being infected with the same clonotype twice. For example, if one group does more travel than another group, then they may be more likely to be infected with different clonotypes. This diversity of clonotypes would make an individual look like they had shorter durations of infection compared with a group that travels less and is re-infected over and over with the same clonotype. i.e. Women who travelled more may only look like they have shorter infection because they are not re-infected with the same clone as much as the men who travelled less. To test this, the authors should look at the distribution of clonotype frequencies and diversity between men and women (e.g. do women tend to have infection with a greater diversity in clonotypes than men?). This might also make men look like they have a lower FOI than they really do, and women a higher FOI than they really do, and it may make it seem like the two groups have the same FOI when actually they are very different.

The reviewer raises an important point. However, there was actually a slightly greater diversity of clones in men, with 34 unique clones found in men and 26 in women. Furthermore, only women over the age of 15 traveled more than males, so this is unlikely to distort the overall findings as proposed.

c) The issue of 4 negatives being required for "clearance" may explain why the infection present in individuals at enrollment is longer than "new infection". This result would change if the end of an infection was defined as the first negative rather than requiring 4 negatives. This may indicate a bias in the analysis towards counting things as one infection that are actually reinfection with the same clonotype.

This was addressed above.

3) The authors should confirm that there is no difference in FOI between the sexes. Even though not significant, there appears to be a trend towards higher FOI in men compared with women in all three age groups (Table 2). This is important because it is not clear how different the FOI in men and women would need to be in order for it to manifest in the differences in prevalence that exist between men and women. It may be that there is really a difference in prevalence observed and that this study does not have statistical power to detect such a difference. Also, calculating FOI as the authors have done requires knowing "new infections" as opposed to just the carriage of old infection, and the definition of what is a new infection is going to influence these conclusions. The authors should further validate this conclusion from their data by: (a) changing their definition of new infection (as in point 2) and recalculating the FOI and determining if there is any significant difference between the sexes. (b) performing a time-to-next-infection analysis (survival analysis, e.g. cox regression), rather than calculate the FOI as new infections over person time. The latter would involve looking at the time until the next new clone appears in each person in a survival model (with appropriate censoring). The question would then be "is sex a significant covariate in a survival model of time to next infection (after including other behavioural factors as well, such as travel)?". If these two other approaches also reveal no difference in FOI between the sexes, then the conclusion that there is no difference in infection pressure on the sexes would be more robust/convincing.

We have performed a sensitivity analysis as discussed in Response #8. Performing a time-to-next-infection analysis as suggested by the reviewer, adjusting for sex and age category and clustering for random effects by participant, results in a HR of new infections in males to females of 1.26 (0.64 – 2.46) and 1.12 (0.76-1.65) by clone. As above, though there is a trend toward more new infections in men, there is no statistical power to detect a difference in this cohort. Since these results are merely another way of analyzing the same data as in the current manuscript and show the same result, these results were not added to the manuscript.

[Editors' note: further revisions were suggested prior to acceptance, as described below.]

The manuscript has been improved but there are some remaining issues that need to be addressed before acceptance, as outlined below:1) The information on baseline infections, a potential confounder to these analyses, still needs further clarification.Subsection “Cohort participants and *P. falciparum* infections”: "25 participants.… were enrolled after initial enrollment…" Please state whether their baseline data contributed to analyses of baseline infections. It seems so from subsection “Data analysis” but should be stated explicitly.

Yes, their baseline data also contributed to the analyses of baseline infections because any infection detected in the first 60 days after their enrollment (whenever that enrollment was) would still be a baseline infection. This is now clarified in the manuscript as follows:

“Data from the 25 dynamically enrolled participants contributed to all analyses, including that of baseline infections.”

Subsection “Cohort participants and *P. falciparum* infections” paragraph two: "35 samples had very low density infections (< 1 parasite/μL) that could not be genotyped and had infections characterized at the event level only". Did these come from 35 unique individuals who had a single infection? And do these represent the difference between 149 individuals with infections versus 114 individuals who had 822 typable samples? If so, please state this explicitly. It appears so from Figure 1.

Yes, they did come from 35 unique individuals. This has been clarified in the manuscript as follows:

“114 participants had 822 successfully genotyped samples and had infections characterized by clone and by infection event. 35 samples (from 35 unique participants) had very low-density infections (< 1 parasite/µL) that could not be genotyped; these infections were characterized at the infection event level only. We achieved a read count of >10,000 for 92% of genotyped samples, identifying 45 unique AMA-1 clones in our population (frequencies and sequences in Supplementary file 1E).”

"117/185.… baseline infections in males". Please state the proportion of all infections included in the analysis that were baseline infections, for males and females separately, at the clone and the infection event level. Baseline infections are an important confounder in this study because they differ from other infections and disproportionately occur in males at the clone level.

I have changed the denominators so that the proportion of baseline infections out of all infections in males vs females is now stated as follows:

“At the clone level, the proportion of baseline infections out of all infections in males was 117/171 (68.4%), compared to 68/116 (58.8%) baseline infections in females (p = 0.10). At the infection event level, there were 54/104 (51.9%) baseline infections in males and 45/89 (50.6%) baseline infections in females (p = 0.89).”

Subsection “Behavioral malaria risk factors and measures of malaria burden”: "differences in prevalence…" Prevalence is defined in subsection “Data analysis”, but it might be helpful to use the term "period prevalence" or a simple explanation here, since many non-epidemiology readers may assume that prevalence here refers to a point prevalence.

I have added a line of explanation as follows:

“Microscopic parasite prevalence was defined by the number of smear-positive routine visits over all routine visits. Parasite prevalence by qPCR was defined as the number of qPCR-positive visits over all routine visits. Thus, these measures represent the average prevalence during routine visits.”

This is not period prevalence since the denominator is not number of individuals; it is instead the average parasite prevalence during routine visits.

In the same section please define COI.

This has now been defined at its earliest appearance.

Subsection “Force of infection by age and sex”: "Force of infection…" Did FOI include baseline infections? Not clear from the definition in subsection “Data analysis”. Please state clearly at the first instance of the term. I assume baseline infections are not included in FOI since it is not possible to know when these started.

By definition, FOI is a measure wherein the numerator is only new infections. Therefore baseline infections are not included in FOI, as new infections are defined as infections detected in a participant after day 60. I have attempted to clarify further in the Materials and methods in several places and added the additional line:

“Force of infection (FOI) was defined as the number of new infections, including malaria episodes, divided by person time. Since the start of baseline infections was not observed, baseline infections were not included in calculating FOI.”

Subsection “Rate of clearance of infection and duration of infection by sex”: "68/105 (64.8%) baseline infections in males…"- as above, please state the proportion of all infections included in the analysis that were baseline infections, for males and females separately.

This is now corrected, as follows:

“At the clone level, 105 baseline infections and 53 new infections were included; there was a slightly higher proportion of baseline infections in males (68/99, 68.7%), compared to the proportion of baseline infections in females (37/59, 62.7%) (p = 0.49). At the infection event level, 58 baseline infections and 51 new infections were included and there was no difference in the proportion of baseline infections by sex, with 32/60 (53.3%) baseline infections in males and 26/49 (53.1%) baseline infections in females (p = 1.0).”

Discussion: "stronger effect in adults than in school-aged children, which was not seen in this cohort". Studies in adolescents have shown a strong relationship between control of parasitemia and adrenarche/DHEAS levels. The authors fail to cite the existing literature showing adrenarche and increasing DHEAS levels which correspond to malaria resistance in males and females. Citing these studies will contribute to the findings here, inasmuch as adrenarche starts earlier in females than males, and the relationship of DHEAS to malaria resistance carries over an extended age window.

I have cited this literature now in the Discussion with its relevance to our study.

Discussion: "a unique area with previously very high transmission intensity that has been greatly reduced in recent years by repeated rounds of IRS." This is not unique. The Garki project provides a similar example. In the Garki project report (p 155), there is no prominent difference in males vs females before the intervention; further, there is a prominent difference after the intervention in the intervention communities, but not in the control communities. This seems relevant to the current study and should be discussed. The Garki project is specifically called out in paragraph three of the Discussion.

By the word “unique” I had meant that it is a specific type of setting, one in which transmission was previously very high and is now very low. I have changed the word “unique” to “specific setting”. Thank you for pointing out additional findings in the Garki study. I had missed this subtlety regarding widening of the prevalence gap between males and females after control measures, and have now incorporated them in the Discussion as follows. It is very interesting that we may be seeing a similar phenomenon.

“Very few studies have been conducted to explore immunological differences between males and females in their response to the malaria parasite. […] More studies are needed to elucidate the relationship between sex-based biological differences between males and females and their impact on the development of effective antimalarial immunity in humans.”

Subsection “Data analysis”: "Infections were censored…" Censored for the clearance analysis only, or censored for both FOI and clearance analysis? Please state clearly.

Censored for the clearance analysis only, as now clarified in the Materials and methods as follows:

“Infections were censored for this analysis if they were only observed in the first three months or the last three months (before January 01, 2018 and after January 01, 2019) because they were not observed for long enough to determine whether clearance occurred.”

Table 1: "Episodes of malaria**, (incidence*)" should just be "Episodes of malaria**", correct?

This has been corrected, thank you for noticing.

2) The authors point out that there is a low likelihood of reinfection with the same genotype. Addressing the issue of clonotype distribution within and between households may not be possible to include easily in this study. For that reason, please consider adding a note of this limitation to the Discussion, since your analysis of the duration of infections (which relies on time infected with the same clonotype) may be impacted if there were a strong tendency of individuals to be reinfected with the same clones.

This is now addressed as follows:

“There were not enough infections to perform a rigorous analysis of the distribution of clones within and between households, but given that the overall force of infection was quite low, the probability of re-infection with the same clone already present in a participant from another member in the household (which could bias toward longer duration of infection) was low and unlikely to have introduced any significant bias by sex.”

3) Please consider adding the following:a) The literature appears conflicting, with significant confounding factors (such as treatment seeking) and study design. There may also be a species difference. See Tjitra et al. PlosMed 2008 Figure 2.

We have added an acknowledgment that some studies do not show a male bias in prevalence or incidence of malaria in the Discussion. We did not choose to add the Tjitra et al., 2008 reference, as Figure 2 is specifically referring to severe malaria and we are not addressing the severity of malaria in females versus males in this paper, only uncomplicated malaria.

“Though there are some conflicting reports in the literature, the majority of studies of malaria incidence and/or prevalence that evaluated associations with sex in late childhood, adolescence and adulthood have found a male bias in the observed measure of burden^10–16^. We note that this is more often observed in hypoendemic settings and may be confounded by factors such as treatment-seeking behavior; however, this male bias has been reported in studies of both *P. vivax* and *P. falciparum*. Overall, these studies consistently suggest that males exhibit higher incidence and/or prevalence of malaria that begins during late childhood, persisting through puberty and the majority of adulthood (excepting the years when pregnancy puts women at higher risk).”

b) Inherent genetic factors remain a plausible explanation and X-linked disorders would warrant particular scrutiny. One point of correct for G6PD, is that although males are at higher risk of severe deficiency (hemizygotes ~5%), the greatest proportion areactually heterozygous females with intermediate deficiency (15-20%). If intermediate deficiency is sufficient to effect clinical susceptibility to disease then this could be an interesting sub group.

Genetic hemoglobinopathies (which would include G6PD) are an unmeasured confounder acknowledged in the Discussion. We realize this is a limitation and will plan to generate data on genetic hemoglobinopathies in future analyses looking at sex-based differences in malaria infection.